# Predicted Cellular Interactors of the Endogenous Retrovirus-K Integrase Enzyme

**DOI:** 10.3390/microorganisms9071509

**Published:** 2021-07-14

**Authors:** Ilena Benoit, Signy Brownell, Renée N. Douville

**Affiliations:** 1Department of Biology, University of Winnipeg, 599 Portage Avenue, Winnipeg, MB R3B 2G3, Canada; benoit-i@webmail.uwinnipeg.ca (I.B.); brownell-s@webmail.uwinnipeg.ca (S.B.); 2Department of Immunology, University of Manitoba, 750 McDermot Avenue, Winnipeg, MB R3E 0T5, Canada

**Keywords:** endogenous retrovirus, integrase, interactome, eukaryotic linear motif, DNA damage response, viral carcinogenesis, cancer, amyotrophic lateral sclerosis, diabetes, model organisms

## Abstract

Integrase (IN) enzymes are found in all retroviruses and are crucial in the retroviral integration process. Many studies have revealed how exogenous IN enzymes, such as the human immunodeficiency virus (HIV) IN, contribute to altered cellular function. However, the same consideration has not been given to viral IN originating from symbionts within our own DNA. Endogenous retrovirus-K (ERVK) is pathologically associated with neurological and inflammatory diseases along with several cancers. The ERVK IN interactome is unknown, and the question of how conserved the ERVK IN protein–protein interaction motifs are as compared to other retroviral integrases is addressed in this paper. The ERVK IN protein sequence was analyzed using the Eukaryotic Linear Motif (ELM) database, and the results are compared to ELMs of other betaretroviral INs and similar eukaryotic INs. A list of putative ERVK IN cellular protein interactors was curated from the ELM list and submitted for STRING analysis to generate an ERVK IN interactome. KEGG analysis was used to identify key pathways potentially influenced by ERVK IN. It was determined that the ERVK IN potentially interacts with cellular proteins involved in the DNA damage response (DDR), cell cycle, immunity, inflammation, cell signaling, selective autophagy, and intracellular trafficking. The most prominent pathway identified was viral carcinogenesis, in addition to select cancers, neurological diseases, and diabetic complications. This potentiates the role of ERVK IN in these pathologies via protein–protein interactions facilitating alterations in key disease pathways.

## 1. Introduction

Viral proteins often usurp and alter cellular signaling pathways. For exogenous viruses, this tweaking of cellular function serves to enhance their replicative success through the modulation of pathways related to virion production, dissemination, cell survival, and immunity [1,2]. It is less clear in what manner ever-present viral symbionts such as endogenous retroviruses (ERVs) interact with the proteome of their hosts.

The genomes of eukaryotic organisms are widely populated with ERVs [3,4,5]. Endogenous retrovirus-K (ERVK/HERV-K) is a biomedically-relevant symbiont within the primate lineage [6,7]. Its expression has been associated with a variety of cancers [8], neurological conditions [9,10,11,12,13], autoimmune diseases [14], and infections [13,15]. A common thread that weaves through ERVK-associated disease is genomic instability. DNA damage and genomic alterations are hallmarks of many cancers [16], as well as in the neurons of patients with the motor neuron disease ALS [17,18]. One protein known to cause DNA damage during the retroviral life cycle is the integrase (IN) enzyme [19]. Recovery from IN-driven lesions is reliant on the host DNA damage response (DDR) [20,21].

We have previously shown that several ERVK insertions in the human genome have the potential to produce functional ERVK IN enzymes with the identical DDE active site motif found in human immunodeficiency virus (HIV) IN [22]. Based on homology modeling, we predict that the ERVK IN enzyme contains all the essential motifs and domain structures for retroviral IN function [22]. A recombinant ERVK-10 integrase enzyme also confirms that it has the potential for strand-transfer activity [23]. A remaining question is how ERVK IN interacts with cellular proteins and pathways, as has been shown for many other retroviral integrases [19,24,25,26].

Retroviral INs are involved in pre-integration complex (PIC) transport [27], viral genome integration into host DNA [19], and virion maturation [28]. Thus, retroviral integrase enzymes exhibit a diversity of cellular partners and have been shown to impact cell signaling and survival processes, including the DDR [19,29]. For example, retroviral IN often recruits viral proteins (reverse transcriptase, matrix, and capsid) and cellular factors (BANF1, HGMA1, LEDGF) to participate in the viral DNA integration process [30,31]. Moreover, successful viral DNA integration requires engagement of the host DDR proteins to repair residual single-stranded DNA gaps flanking the integration site [20,29,32]. In contrast, failed provirus insertion or unresolved lesions can lead to double-stranded DNA (dsDNA) breaks in the host genome [29]. The level of γH2AX foci is positively correlated with the number of double-stranded DNA breaks (DSB) in mammalian cells, and it is widely used as a quantitative biomarker of retrovirus-mediated DSBs [19,33,34]. This genomic damage is particularly hazardous to the cell, as DSB potentially lead to chromosomal rearrangements, cellular deregulation, and apoptosis [35,36]. Thus, as an intrinsic protective measure, select host proteins (RAD51, Kap1, TREX1, p21, HDAC10, TRIM33) are known antagonists of the retroviral integration process [31,37,38,39,40]. Many studies have identified direct protein binding partners and cellular complexes which interact with HIV integrase [41,42,43]; in contrast, the ERVK IN interactome remains unknown.

A complicating factor for the development of model systems to study the impact of ERVK proteins in vivo is that many other organisms contain ERVs with similarity to ERVK. Given the known cellular impacts of retroviral integrases, we hypothesized that a computational biology approach would identify potential cellular partners of ERVK IN and point toward its capacity to modulate cellular pathways. Additionally, a comparison with similar integrases in eukaryotic organism and model species may inform the future establishment of in vivo models for ERVK IN-driven pathology.

## 2. Materials and Methods

### 2.1. Database Curation

Integrases with sequence similarity to ERVK IN (based on ERVK-10 [22]; P10266.2) were identified using the National Centre for Biotechnology’s (NCBI) Protein–protein Basic Local Alignment Search Tool (BLASTp) within the non-redundant (nr), model organisms (mo), and transcriptome shotgun assembly proteins (tsa) databases [44]. Default algorithm parameters were used, with E-value cut-offs for each database as follows: E < 3.0 × 10^−70^ (nr), E < 0.01 (mo), E < 2.0 × 10^−10^ (tsa). Sequences were grouped based on phylogeny as informed by ICTV (International Committee on Taxonomy of Viruses; 2021, https://talk.ictvonline.org/ (accessed on 16 May 2021)) or OneZoom (OneZoom Tree of Life Explorer; version 3.4.1; Software for Technical Computation; United Kingdom, 2021, https://www.onezoom.org/ (accessed on 16 May 2021)) [45] and are listed in Table A1, Table A2, Table A3 and Table A4.

### 2.2. Protein Alignments and Eukaryotic Linear Motif Annotation

The ERVK IN protein sequence, as well as select representative integrases from exogenous Betaretroviruses (Figure 1) or endogenous retroviruses (Figure 2) were aligned using Geneious Prime (version 2021.0.3; Software for Technical Computation; San Diego, CA, USA; Auckland, New Zealand, 2021) software [46]. A global alignment with free end gaps using BLOSUM62 matrix was performed. Longer sequences were truncated to overlap with the ERVK IN reference sequence. Figures depict the sequence logo and integrase active sites, with HHCC and DDE regions highlighted based on Conserved Domains Database (CDD) annotation [47].

Each aligned integrase sequence was submitted to the Eukaryotic Linear Motif (ELM; Software for Technical Computation; 2020, http://elm.eu.org/ (accessed on 16 May 2021)) resource [48]. A complete listing of ELMs identified in each integrase is presented in Table 1 and Table 2. ELMs unique to ERVK IN, as well as ELM sites exhibiting motif consensus above 70% with other integrases, were annotated in Figure 1 and Figure 2.

### 2.3. STRING Analysis and KEGG Pathways

To identify potential ERVK IN binding partners based on ELM motifs, the names of interacting proteins were curated from each ELM reference page. When only a general interaction domain for a given ELM was listed, it was further linked to the InterPro database to curate a list of human proteins containing the interaction domain. Based on the 48 ELMs identified in ERVK IN, a total of 213 putative human protein interaction partners were identified (Table A5). The list was submitted to STRING (String Consortium; version 11.0; Software for Technical Computation; 2020, https://string-db.org/, accessed on 16 May 2021) for network analysis. Full network analysis was performed using Experiment and Databases as active interaction sources. Nodes indicate submitted query proteins only, with edges indicating confidence lines with a minimum interaction score of 0.9 (highest confidence). Query proteins unlinked to the network were excluded from analysis. A payload list was used to color hub proteins based on cellular function. KEGG pathways associated with the network analysis (E value < 1.0 × 10^−5^) were presented in a heatmap using GraphPad Prism (version 9.1.1) software, and the full list of KEGG pathways is presented in Table A6.

## 3. Results

### 3.1. Characterization of Eukaryotic Linear Motifs in ERVK Integrase and Other Betaretroviral Integrases

To establish which exogenous and endogenous retroviruses contain integrase sequences most similar to ERVK IN, we performed BLASTp searches using the nr, mo, and tsa NCIB databases. As expected, exogenous Betaretroviruses were identified through BLASPp search, which included multiple hits for Mouse mammary tumor virus (MMTV), Mason–Pfizer monkey virus (M-PMV), Enzootic Nasal Tumor Virus (ENTV), and Jaagsiekte sheep retrovirus (JSRV) (Table A1).

Eukaryotic linear motif (ELM) analysis of a representative sequence from each genus was compared with ERVK IN and revealed the conservation of select protein motifs (Table 1, Figure 1). Apart from the conservation of the HHCC region and DDE active site motif, all betaretroviral integrases also contained many interaction motifs related to DRR, including Pin1 via [ST]P WW domain interaction motifs [49], PP1c docking motif for target dephosphorylation [50], and a S-X-X-S/T CK1 phosphorylation site [51]. All INs except for MMTV contained a low-affinity BRCA1 carboxy-terminal BRCT domain binding motif (CSKAF, aa. 126–132). Betaretroviral INs were also predicted to be phosphorylated by the cell cycle checkpoint kinases NEK2 [52] and PLK-1 [53], as well as interact with a canonical arginine-containing phospho-motif within cell cycle regulating 14-3-3 proteins [54]. Numerous cell signaling protein interactions were predicted including YXXQ motifs for the SH2 domain binding of STAT3 [55], additional SH2 binding motifs related to STAT5 [56] and SRC family kinases [57], SH3 binding motifs with non-canonical class I recognition specificity [58], an IAP-binding motif (aa. 1–4) for interaction with inhibitor of apoptosis proteins (IAP) [59], an ITIM motif [60], several GSK3 phosphorylation sites [61], and proline-directed ERK/p38 MAPK phosphorylation sites [62]. In addition, most betaretroviral IN enzymes contained features related to protein trafficking, such as a Wxxx[FY] motif (aa. 133–137 in ERVK, MMTV, ENTV, and JRSV) that binds Pex13 and Pex14 for peroxisomal import [63], a SxIP motif (aa. 137–150) that binds to EBH domains in end-binding proteins involved in microtubule transport [64], and a tyrosine-based YXXØ sorting signal (aa. 75–78) for interaction with the μ-subunit of adaptor protein complex [65] and a PEXEL-like motif [66]. The DDE region displayed the most consistent pattern of conserved ELMs among the betaretroviral INs. It is important to note that despite the similar complement of ELM motifs in betaretroviral integrases, many were positioned at sites differently than in ERVK IN. Additional ELMs and their motif frequencies in individual betaretroviral integrases are listed in Table 1.

### 3.2. Characterization of Eukaryotic Linear Motifs in ERVK Integrase and Other Endogenous Integrases

ERVK integrase-like sequences were found in boreoeutherians, including the Euarchontoglires (primates, rodents, and pikas), and Laurasiatherians (ungulates), along with other clades including Euteleostomi (birds) and Protostomes (worms, insects, and water fleas) (Table A2, Table A3 and Table A4). Results ranged from 26.43 to 83.77% identity and E values ranged from 0.001 to 2.0 × 10^−127^, demonstrating a high degree of similarity with ERVK IN.

The conservation of ELM motifs was apparent (Table 2, Figure 2), including DDR-related canonical 14-3-3 interaction motifs and WDR5 interaction, cell signaling associated with USP7 binding, IAP-binding motif, STAT5 binding motifs, SH3 protein interaction, as well as phosphorylation sites for CK proteins, GSK3, NEK2, polo-like kinases, and p38. Many LIR motifs for engaging Atg8 proteins during selective autophagy were also apparent. Finally, all IN displayed Pex14 binding motifs and potential to interact with the μ-subunit of the adaptor protein complex. Additional ELMs and their motif frequencies in individual endogenous ERVK-like INs are listed in Table 2.

ELMs within endogenous IN but not or rarely identified in ERVK IN were also noted. An Apicomplexa-specific variant of the canonical LIR motif that binds to Atg8 protein family members was present in all endogenous INs except for ERVK IN. In addition, WDR5 binding motifs were much more prevalent in endogenous INs (5–12 sites) other than ERVK IN (only two sites). ERVK IN contained a single MAPK docking site for ERK/p38, whereas other endogenous INs contained several other motifs for MAPK interaction. Lastly, only human and macaque ERVK INs displayed high-affinity BRCT domain interaction motifs.

### 3.3. Unlike Similar Enzymes, the ERVK Integrase Contains Distinct ELM Signatures

Two motifs in ERVK IN stand out as unique to this virus, while other signatures are enriched in ERVK IN as compared with similar integrases.

#### 3.3.1. ERVK Integrase has a High-Affinity BRCA1 Binding Site

Among all the integrases examined, only ERVK IN in human and macaque harbored a high-affinity binding site for the BRCT domain of BRCA1 (aa. 125–131, CSKAFQK) (Table 1 and Table 2, Figure 1 and Figure 2).

#### 3.3.2. ERVK Integrase C-Terminus Contains a 14-3-3 Binding RASTE Motif

Although 14-3-3 protein binding was predicted as conserved among ERVK-like integrases, only ERVK IN contained a C-terminal RASTE motif (aa. 276–280) mediating strong interaction with 14-3-3 proteins (Table 1, Figure 1). This suggests a putative ERVK IN interaction with 14-3-3 proteins through both canonical phospho-sites and a C-terminal phospho-site.

#### 3.3.3. ERVK Integrase Is Likely Post-Translationally Sumoylated

Unlike all other INs examined, only ERVK and MMTV contain a C-terminal inverted version (D/ExKphi) of the canonical sumoylation motif [67]. Considering that sumoylation often causes re-localization of nuclear proteins, this modification may be related to ERVK IN nuclear positioning, association with chromatin, and ultimately successful integration of viral DNA [68,69].

#### 3.3.4. ERVK Integrase Exhibits Enhanced Interaction Potential with DDR Proteins

Phospho-Ser/Thr binding domain proteins are key hub proteins in cell cycle regulation and DDR, and they include 14-3-3 proteins, WW domains, Polo-box domains, WD40 repeats, BRCA1 carboxy-terminal (BRCT) domains, and Forkhead-associated (FHA) domains [54], all of which are interacting domains of ELMs identified in ERVK IN (Table 1 and Table 2, Figure 1). Additionally, ERVK IN contained five (ST)Q motifs, which are potential phosphorylation sites for PIKK proteins, such as DDR-related proteins ATR, ATM, DNA-PK, and multi-functional protein mTOR [70]. As compared with exogenous betaretroviruses and endogenous ERVK-like integrases, ERVK IN displayed a greater number of DDR-related motifs: FHA domain protein interaction sites (6), PLK-1 phosphorylation sites (4), and PP1c docking motif for target dephosphorylation (3) [50]. In contrast to MMTV, ENTV, JSRV, and most other endogenous integrases, fewer WD40 repeat domain WDR5 interaction sites were found in ERVK IN (2 vs. 5–12 sites each). This suggests ERVK IN has potentially shifted away from WDR5 interaction in favor of BRCA1 (or BRCT domain) interaction as a means to interact with the DDR pathway [54,71].

#### 3.3.5. ERVK Integrase Contains Canonical Selective Autophagy Motifs

Unlike any of the exogenous betaretroviruses, only ERVK IN and some endogenous integrases contained canonical LIR motifs (ELME000368) for binding Atg8 protein family members (Table 1 and Table 2, Figure 2). All endogenous INs contained nematode-specific LIR motifs (ELME000370). Additionally, most endogenous INs housed Apicomplexa-specific LIR motifs (ELME000369), whereas ERVK IN did not.

### 3.4. ERVK Interactome Reveals Association with a Diversity of Cellular Pathways

Based on ELMs identified in ERVK IN, a curated list of potential interacting proteins was generated and used to build an ERVK IN interactome network using STRING software (Figure 3). The ERVK IN network contained 189 nodes and 692 edges (expected number of edges 222), resulting in a significant PPI enrichment (*p* < 1.0 × 10^−16^). Only direct interactor query proteins are shown without links to second shell interactions. To illustrate key nodes and hub proteins, select network proteins were colored based on function related to DDR, cell cycle, apoptosis, cell signaling, or cellular transport. A complete list of the KEGG pathways significantly associated with the network is presented in Table A6.

#### 3.4.1. Many DNA Damage Response Proteins Are Potential ERVK Integrase Interactors

Gene ontology (GO) biological processes that were significantly enriched in the network included cellular response to DNA damage stimulus (*p* < 4.4 × 10^−18^), DNA repair (*p* < 4.24 × 10^−12^), DNA damage checkpoint (*p* < 4.89 × 10^−12^), double-strand break repair via non-homologous end joining (NHEJ) (*p* < 3.57 × 10^−9^), double-strand break repair (*p* < 1.06 × 10^−8^), and signal transduction in response to DNA damage (*p* < 2.12 × 10^−8^). Select BRCT domain containing proteins emerged as nodes with a higher-than-average degree of connections, including BRCA1, BARD1, NBN, MDC1, RCF1, TOPBP1, TP53BP1, and PAXIP1, while PARP1 and DRKDC (DNA-PK) appear to be hub proteins between DDR and apoptosis. The ERVK IN network also displayed four prominent DDR-related FHA proteins: CHEK2, NBN, MDC1, and RNF8.

#### 3.4.2. ERVK Integrase Likely Modulates Cell Cycle Pathways

GO biological processes that were significantly enriched in the network included regulation of cell cycle (*p* < 1.45 × 10^−33^), cell division (*p* < 1.12 × 10^−20^), regulation of cyclin-dependent serine/threonine kinase activity (*p* < 6.86 × 10^−20^), mitotic cell cycle (*p* < 2.78 × 10^−17^), regulation of apoptotic process (*p* < 1.45 × 10^−17^), and cell cycle checkpoint (*p* < 6.72 × 10^−12^). Many cyclins and 14-3-3 proteins were identified in the network and are listed in Table A5. IAP-containing protein BIRC5 (also known as survivin) was also identified, which is suggestive of negative regulation of programmed cell death pathways [72]. PLK1 and NEK2 were also tied into the cell cycle framework and are both regulators of mitosis, in addition to displaying oncogenic properties [73,74].

#### 3.4.3. Cell Signaling Pathways Associated with the ERVK Interactome

Among the potential signaling pathways often targeted by retroviruses, ERVK IN-associated cascades emerged as Forkhead box O (FoxO) signaling [75], p53 signaling [76], ErbB signaling [77], Wnt signaling [78], modulation of kinase activity [79,80], and multiple aspects of immune signaling [81] (Figure 4). Within these pathways, prominent immune-related signaling intermediates included STAT3 [55], STAT5 [56], and TRAF2 [82]. The SH2 and SH3 containing tyrosine-protein kinase ABL1 (Abelson murine leukemia viral oncogene homolog 1 [57]) appears to be a key hub protein linking DDR and downstream signaling cascades.

#### 3.4.4. ERVK Integrase May Utilize Specific Cellular Transport Systems

The ERVK interactome contains proteins related to cellular transport. EB1 (MAPRE1) is an end-binding (EB) protein connected with both cell cycle and signaling pathways and is functionally associated with the regulation of microtubule dynamics [83]. Adaptor protein complex 2 associated proteins (AP2M1 and CTTN) were identified and indicate a role in cargo internalization via clathrin-mediated endocytosis and actin dynamics [65,84]. Lastly, ERVK IN may interact with Pex14 and Pex13 independently of the main network for peroxisome import [63]. While these pathways were likely important for the ancestral exogenous ERVK to transverse the cell and mediate infection, it remains unclear how endogenous IN may interact with these systems.

### 3.5. Diseases and Pathways Implicated in the ERVK Integrase Interactome

#### 3.5.1. Cancers

Viral carcinogenesis was the top KEGG pathway identified in the ERVK IN network analysis (strength 1.22, E value 3.7 × 10^−23^), with 29 of 183 proteins represented (Figure 4). KEGG pathways for several known ERVK-associated cancers were also identified, including lung cancer [85], myeloid leukemia [86], and hepatocellular carcinoma [87] (Figure 4). Glioma was also identified, yet ERVK is downregulated in this condition [88]. Aligned with cellular transformation, proteins associated with cell cycle were also over-represented in the pathway analysis, which are specifically related to the cyclin docking site ELM (DOC_CYCLIN_RxL_1) and numerous FHA domain protein interaction sites (LIG_FHA_1 and LIG_FHA_2) in ERVK IN (Table 1 and Table 2, Figure 1 and Figure 2).

#### 3.5.2. Neurological Disease

KEGG pathways for several ERVK-associated neurological conditions were identified, including ALS [9,12], Alzheimer’s disease [89], and prion disease [90] (Figure 4). Specifically, long-term potentiation and synaptic neurotransmitter release (dopaminergic, glutamatergic, cholinergic, serotonergic, and GABAergic) were associated with the ERVK IN interactome.

#### 3.5.3. Diabetes

The role of ERVK in diabetes remains contentious [91,92,93,94]. However, network analysis suggests that the ERVK IN interactome is potentially linked to AGE-RAGE signaling in diabetic complications, insulin signaling, and insulin resistance (Figure 4).

## 4. Discussion

ERVK expression has been repeatedly associated with human disease states including cancer, neurological disease, and diabetes. By exploring the potential ERVK integrase interactome, we can postulate how this viral symbiont may contribute to disease pathogenesis via interaction with key proteins and pathways. Our analysis reveals that viral carcinogenesis and modulation of the DNA damage response are the most likely pathways to be pathologically associated with ERVK IN expression.

Retroviral integrase activity causes DNA lesions in the host genome as part of the proviral integration process [19]; therefore, interactions with DDR pathways are to be expected. Several DDR proteins have been shown to be essential for provirus suture into the host genome and maintenance of genome fidelity [19]. Yet, the impairment of select aspects of DDR has also been documented in exogenous retroviral infections, including HIV [95,96] and HTLV-1 [97,98]. This may be driven by the fact that NHEJ proteins also play an essential role in innate immune recognition of retroviral cytosolic ssDNA intermediates and dsDNA pre-integration complexes [98,99]. Thus, retroviruses must balance the benefits and drawbacks of DDR outcomes through the engagement and modulation of specific proteins.

BRCT domain, FHA domain, and 14-3-3 proteins work in concert during the DNA damage response (reviewed in [100]). Many of these DDR proteins are also cellular targets of retroviruses and oncogenic viruses [98,101,102,103,104]. BRCA1 BRCT domains recognize phosphopeptides based on a pSXXF motif, but XX residues and the surrounding amino acids also impact binding affinity and selectivity [105]. All the betaretroviral INs examined showed the capacity to interact with BRCT domains. However, only ERVK IN displayed a high affinity (S.F.K) BRCA1 BRCT domain binding site; the only other similar ELM structure is found in the DDR protein Fanconi anemia group J protein (FACJ/BACH1) [106]. It is also possible that dual anchoring onto the ERVK IN using both a BRCT domain and an FHA domain found in NBN or MDC1 may strengthen protein–protein interactions.

The utilization and evasion of 14-3-3 proteins are common among many viruses [104]. ERVK IN is unique in having a C-terminal RASTE motif, in addition to two other canonical arginine containing phospho-motifs recognized by 14-3-3 proteins. Given that an elevated expression of 14-3-3 proteins occurs in both cancers and neurodegenerative diseases [107,108], ERVK IN interaction with 14-3-3 protein members may be related to either modulation of the cell cycle and oncogenesis or regulation of protein aggregation, respectively. The deregulation of 14-3-3 and RAF kinase interaction can also lead to inappropriate downstream MAPK activity (associated with oncogenesis) [54,109] and may be an aspect to consider for the predicted ERVK IN network.

ABL1 appears to be a key hub protein linking DDR and downstream signaling cascades. Interestingly, DDR is known to be a rapid driver of ABL1 activation [110]. The ablation of ABL1 reduces retrovirus integration [111,112], while active ABL1 can turn on the HIV promoter independently of HIV Tat [113]. Putative interaction between ERVK IN and ABL1 may have been important for ERVK integration into germline cells, and it may additionally play a role in ERVK expression, specifically in neurodegenerative disease displaying enhanced ABL1 activity [114,115].

DDR is intimately tied to innate immune response, specifically NF-κB activation [116]. Considering ERVK’s dependence on NF-κB for driving its own expression [11], it is conceivable that ERVK IN plays a role in preparing the host cell for viral transcription. 14-3-3ϵ activity is key in driving ATM-TAK1-mediated NF-κB signaling during DDR [117,118]; thus, the predicted ERVK IN interaction with 14-3-3ϵ (YWHAE) may be a mechanism to favor viral transcription. The MAPK p38 was also predicted to both phosphorylate and bind ERVK IN. This association may be linked to p38′s regulation of inflammatory signaling, as well as its capacity to enhance the transcriptional activity of NF-κB p65 via modulation of the acetyltransferase activity of coactivator p300 [119]. Sustained NF-κB activity is linked to oncogenesis [116] and ties into the strongest ERVK IN-linked KEGG pathway: viral carcinogenesis. However, enhanced ERVK IN-associated NF-κB signaling may also fit with inflammatory and neurodegenerative conditions.

ERVK IN stability and protein turnover is likely linked to its cellular protein partnerships. In the case of HIV, binding select cellular proteins such as LEDGF/p75 and Ku70 prevents integrase proteosomal degradation [120,121]. Similarly, c-Jun N-terminal kinase (JNK) S^57^ phosphorylation of the core domain can make HIV IN a target for Pin1, thus enhancing its stability and activity [122,123]. In this study, Pin1′s WW domain was predicted to be an interactor based on three [ST]P motifs in the C-terminal portion of ERVK IN. This raises the possibility that similar to many other viral proteins [124], ERVK IN may be stabilized through Pin1 interaction. The functional significance of this interaction may underlie how elevated levels of ERVK IN are maintained and potentially drive pathology in select diseases, such as ALS and cancer.

Distinct from other exogenous betaretroviruses, only ERVK IN and some endogenous integrases contained canonical LIR motifs for binding Atg8 protein family members. Mammalian Atg8-like proteins include LC3 and GABARAP families, which mediate selective autophagy, as well as play essential roles in antiviral defense and innate immune signaling [125]. However, it is often observed that viruses subvert autophagy processes to avoid viral protein clearance and repurpose Atg8 proteins as well as autophagosomal membranes for viral replication [125,126]. Considering the perturbances of autophagy in neurodegenerative disease [127,128], the interaction between ERVK IN and Atg8 proteins warrants further investigation.

Consistent with genomic instability profiles in cancer [129], ALS [130], and Alzheimer’s disease [131], the ERVK interactome analysis identified each of these conditions as significant KEGG pathways. Despite differences in clinical presentation, the molecular underpinnings in cancer and neurodegenerative disease are remarkably similar and include alterations in DDR [129,130,132], 14-3-3 expression [133,134], p53 signaling [135,136], p38 signaling [137,138], and Wnt signaling [139,140]—which are all KEGG pathways enriched in the ERVK IN network. AGE-RAGE signaling was also identified as a potential pathway associated with the ERVK IN interactome. Not only is this pathway implicated in diabetic complications [141], but it also plays a role in nuclear response to DNA damage [142], carcinogenesis [143], and inflammatory neurodegenerative diseases [144]. Collectively, our results point to ERVK IN driving a pattern of pathology that, depending on cellular context, may lead to carcinogenesis, neurodegeneration, or contribute to diabetic complications. However, the engagement of DDR can also have beneficial impacts on lifespan extension, depending on tissue context and host genotype; thus, non-pathogenic effects of ERVK IN should also be considered [145,146].

Apart from the importance of putative ERVK IN interaction partners, it is also important to consider which cellular proteins were not associated with the ERVK IN interactome. One interaction that was not predicted was with LEDGF/p75, and indeed, this interactor is limited to partnership with lentiviral integrases [147,148]. Another set of DDR proteins commonly found to impact retroviral integration and replication is the DNA-PK complex [99,149]. HIV integrase directly interacts with Ku70 [120]; while ERVK IN was predicted to be phosphorylated by DNA-PKcs (PRKDC), it contained no ELMs to suggest direct interaction with Ku80 or Ku70. Another apparent difference is the use of EB proteins in microtubule trafficking for HIV and ERVK. ERVK IN contained an SxIP motif that binds EBH domains, whereas HIV capsid conversely has EB-like motifs that interact with SxIP motifs in plus-end tracking protein (+TIP) [150]. These genus-specific distinctions are likely to emerge as important considerations for therapeutic targeting strategies and imply that pharmaceuticals geared toward HIV infection may not consistently translate for use in ERVK-associated disease.

Another consideration that stems from this study is the choice and caveats of using animal models in ERVK research. A diversity of animals outside of the primate lineage are host to ERVK IN-like sequences, such as rodents, ungulates, fish, and insects. Drosophila, a common model organism, also contained ERVK IN-like elements in their genome, specifically LTR retrotransposons *flea* and *Xanthias*, as identified by FlyBase (Table A4). The transposable element *Xanthias* is known to be active in *D. melanogaster* [151,152], and it shares a degree of similarity with ERVK IN. The presence and activity of these ERVs is an important factor to consider when performing experiments.

It is shocking how little we understand of the impact endogenous viral symbionts have on cellular functioning. Herein, we have predicted that ERVK IN may participate in the modulation of cellular pathways such as DDR, cell cycle regulation, and kinase signaling cascades by way of select protein interaction motifs. The main caveat of in silico predictions is the requirement for experimental validation; while research into ERVK IN is currently underway, this study suggests there remains a myriad of disease-related betaretroviral integrase interactions to explore.

## Figures and Tables

**Figure 1 microorganisms-09-01509-f001:**
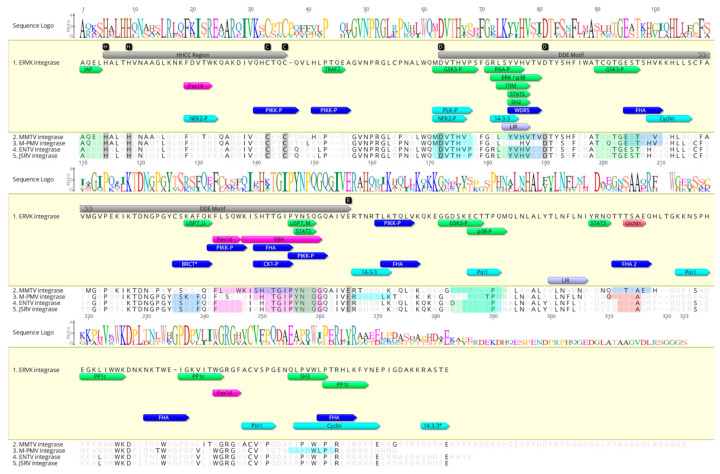
ERVK integrase and exogenous betaretrovirus integrases share common eukaryotic linear motifs. In silico examination of the conserved and differential eukaryotic linear motifs (ELMs) within Endogenous retrovirus-K (ERVK) and similar betaretroviral integrases. A betaretroviral integrase consensus sequence was constructed using GenBank sequences as follows: Endogenous retrovirus-K (ERVK; P10266.2), Exogenous mouse mammary tumor virus (MMTV; AAF31469.1), Mason–Pfizer monkey virus 5 (M-PMV; BBG56792.1), Enzootic nasal tumor virus (ENTV; ANG58699.1) and Jaagsiekte sheep retrovirus (JSRV; NP_041186.1). The HHCC region and DDE active site motif (gray, with key aa. in black) was positioned based on Conserved Domains annotations. ELMs were grouped based on related pathways: DNA damage response (dark blue), cell cycle (cyan), cell signaling (green), cell trafficking (magenta), autophagy (mauve), and glycosylation (red). ELM abbreviations used include: 14-3-3 = 14-3-3 protein interaction site, BRCT = BRCA1 C-terminus domain interaction site, CK1-P = casein kinase 1 phosphorylation site, Cyclin = cyclin docking site, EBH = end binding homology domain interaction site, ERK/p38 = ERK1/2 and p38 MAP kinase docking site, FHA = Forkhead-associated domain interaction site, GlyNH = glycosaminoglycan attachment site, GSK3-P = GSK3 phosphorylation site, IAP = inhibitor of apoptosis protein interaction site, ITIM = immunoreceptor tyrosine-based inhibitory motif, LIR = site that interacts with Atg8 protein family members, NEK2-P = NEK2 phosphorylation site, p38-P = p38 phosphorylation site, Pex14 = peroxisomal membrane docking via Pex14, PIKK-P = PIKK family phosphorylation site, Pin1 = docking site for Pin1 via WW domain interaction, PKA-P = PKA phosphorylation site, PLK-P = polo-like kinase phosphorylation site, PP1c = protein phosphatase 1 catalytic subunit docking motif, SH2 = Src homology 2 domain interaction motif, SH3 = interaction site for non-canonical class I recognition specificity SH3 domains, STAT3 = STAT3 SH2 domain binding motif, STAT5 = STAT5 SH2 domain binding motif, TRAF2 = major TRAF2 binding consensus motif, USP7 = USP7 MATH (M) or UBL2 (U) domain interaction sites, WDR5 = interaction motif for WDR5 via WW domain interaction. Asterisks indicate ELMs unique to ERVK. Sequence alignment and annotation were performed using Geneious Prime software.

**Figure 2 microorganisms-09-01509-f002:**
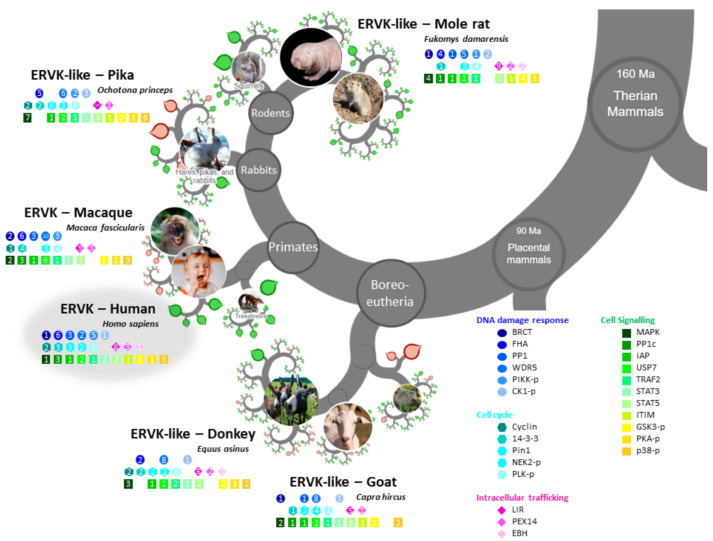
ERVK integrase and similar endogenous integrases share eukaryotic linear motifs patterns. Modified OneZoom image illustrating the conservation of ELM motifs in integrases from eukaryotic organisms (*Homo sapiens, Macaca fasicicularis, Fukomys damarensis, Ochotona princeps, Equis asinus,* and *Capra hircus*). Motifs are color-grouped according to function; DDR (blue), cell cycle (cyan), cell signaling (green), and intracellular trafficking (magenta). The number in each colored shape refers to the number of motifs with the respective integrase enzyme.

**Figure 3 microorganisms-09-01509-f003:**
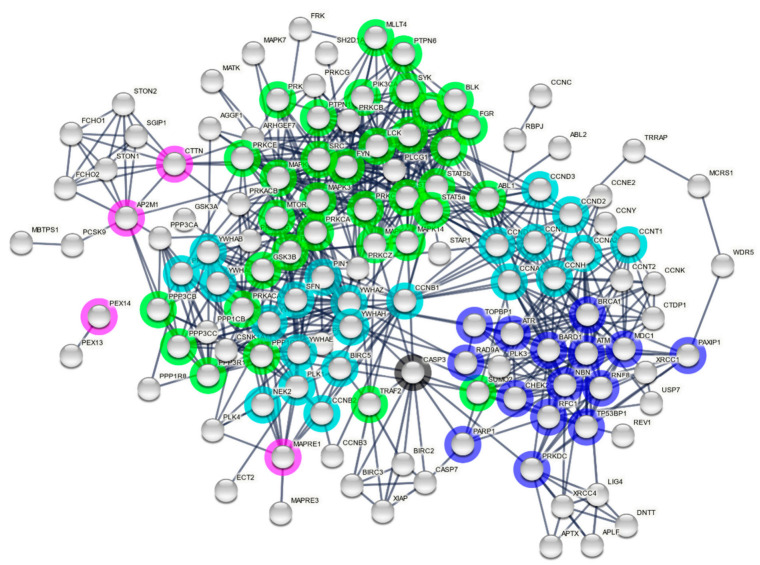
Predicted ERVK integrase interactome. Cellular proteins containing complementary interaction motifs for ELMs identified in Endogenous retrovirus-K (ERVK) integrase were listed as query proteins for STRING network analysis. Only query proteins with a minimum interaction score of 0.9 based on experiments and databases as interaction sources are indicated. Edges indicate both functional and physical protein associations. A payload list was generated to color nodes and hubs related to dominant pathways: DNA damage response (dark blue), cell cycle (cyan), apoptosis (black), cell signaling (green), and cell transport (magenta).

**Figure 4 microorganisms-09-01509-f004:**
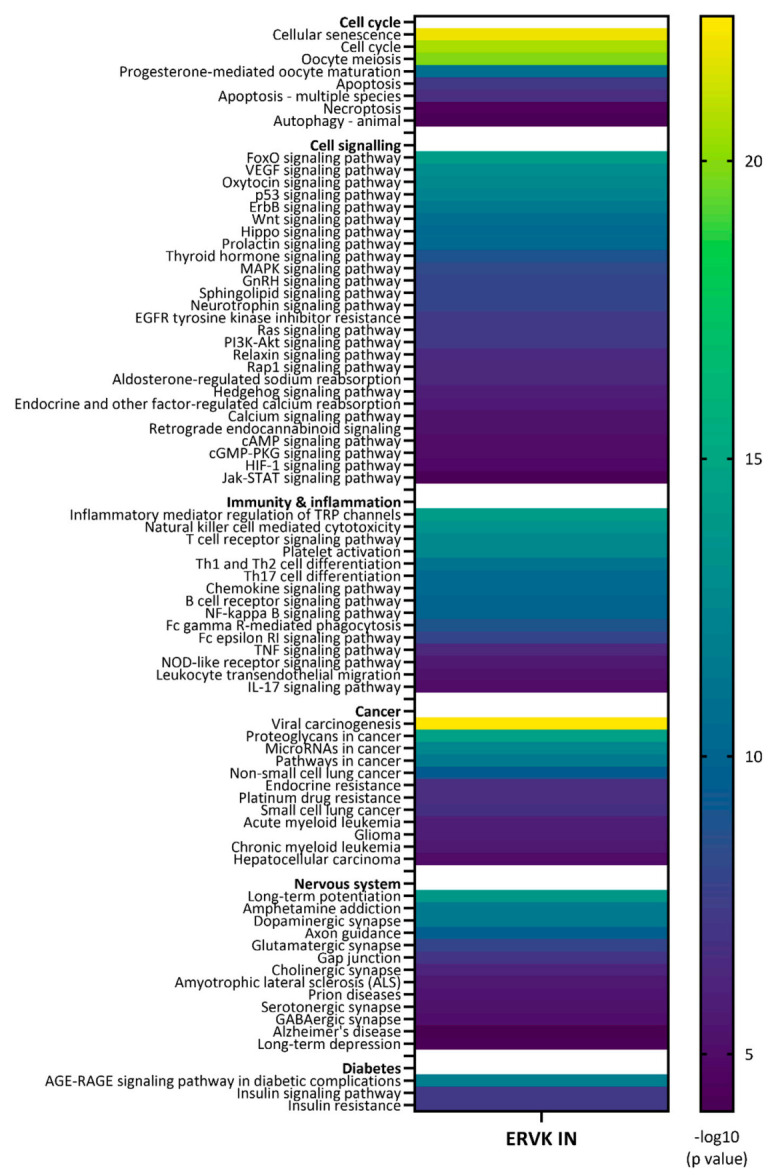
KEGG pathways associated with ERVK integrase interactome. Predicted interacting partners were curated based on ERVK IN ELM motifs and submitted to STRING network analysis software. Enriched KEGG pathways are reported along with significance scores (−log10 *p* value). ERVK IN is predicted to interact with cellular pathways involved in the cell cycle, cell signaling, immunity, and inflammation, as well as disease pathways associated with several cancers, the nervous system, and diabetes.

**Table 1 microorganisms-09-01509-t001:** ELM motifs in integrases from ERVK and exogenous betaretroviruses.

ELM Motif		ELM Accession	Alignment Notation	Integrase					Conservation
				ERVK	MMTV	M-PMV	ENTV	JSRV	
Cleavage and degradation	CLV_C14_Caspase3-7	ELME000321		1	0	0	0	2	0.4
	CLV_PCSK_KEX2_1	ELME000108		1	0	0	0	0	0.2
	CLV_NRD_NRD_1	ELME000102		0	1	0	1	1	0.6
	CLV_PCSK_PC1ET2_1	ELME000100		1	0	0	0	0	0.2
	CLV_PCSK_SKI1_1	ELME000146		5	4	4	1	0	0.8
	DEG_APCC_DBOX_1	ELME000231		0	0	0	1	0	0.2
Docking	DOC_CKS1_1	ELME000358		0	1	0	0	0	0.2
	DOC_CYCLIN_RxL_1	ELME000106	Cyclin	2	0	2	0	0	0.4
	DOC_MAPK_gen_1	ELME000233		0	2	0	1	1	0.6
	DOC_MAPK_MEF2A_6	ELME000432	ERK/p38	1	1	0	1	1	0.8
	DOC_PP1_RVXF_1	ELME000137	PP1c	3	1	1	1	1	1.0
	DOC_PP2B_LxvP_1	ELME000367		1	0	1	0	0	0.4
	DOC_PP4_FxxP_1	ELME000477		0	1	0	1	1	0.6
	DOC_USP7_MATH_1	ELME000239	USP7_M	1	3	0	2	2	0.8
	DOC_USP7_UBL2_3	ELME000394	USP7_U	1	0	1	0	0	0.4
	DOC_WW_Pin1_4	ELME000136	Pin1	3	6	1	3	2	1.0
Ligand	LIG_14-3-3_CanoR_1	ELME000417	14-3-3	2	1	1	2	2	1.0
	LIG_14-3-3_CterR_2	ELME000418	14-3-3 *	1	0	0	0	0	0.2
	LIG_Actin_WH2_2	ELME000313		0	1	0	1	1	0.6
	LIG_APCC_ABBA_1	ELME000435		0	1	0	0	0	0.2
	LIG_BIR_II_1	ELME000285	IAP	1	1	1	1	1	1.0
	LIG_BIR_III_4	ELME000293		0	0	0	1	0	0.2
	LIG_BRCT_BRCA1_1	ELME000197	BRCT	1	0	1	1	1	0.8
	LIG_BRCT_BRCA1_2	ELME000198	BRCT1 *	1	0	0	0	0	0.2
	LIG_CSL_BTD_1	ELME000410		1	1	0	0	0	0.4
	LIG_EH1_1	ELME000148		0	0	1	0	1	0.4
	LIG_eIF4E_1	ELME000317		0	1	0	0	0	0.2
	LIG_FHA_1	ELME000052	FHA	5	1	2	0	1	0.8
	LIG_FHA_2	ELME000220	FHA 2	1	1	1	0	0	0.6
	LIG_LIR_Apic_2	ELME000369		0	3	2	1	1	0.8
	LIG_LIR_Gen_1	ELME000368	LIR	2	0	0	0	0	0.2
	LIG_MAD2	ELME000167		0	0	0	1	0	0.2
	LIG_NRBOX	ELME000045		0	0	0	1	0	0.2
	LIG_LIR_Nem_3	ELME000370		0	6	7	1	4	0.8
	LIG_Pex14_1	ELME000080	Pex14	1	0	0	0	0	0.2
	LIG_Pex14_2	ELME000328	Pex14	2	2	1	1	1	1.0
	LIG_PTB_Apo_2	ELME000122		0	0	1	0	1	0.4
	LIG_PTB_Phospho_1	ELME000095		0	0	0	0	1	0.2
	LIG_RPA_C_Fungi	ELME000382		0	0	1	1	1	0.6
	LIG_SH2_CRK	ELME000458		0	2	0	0	0	0.2
	LIG_SH2_NCK_1	ELME000474		0	1	0	0	0	0.2
	LIG_SH2_PTP2	ELME000083	SH2	1	1	0	1	1	0.8
	LIG_SH2_SRC	ELME000081	SH2	1	1	1	1	1	1.0
	LIG_SH2_STAP1	ELME000465		2	1	0	0	0	0.4
	LIG_SH2_STAT3	ELME000163	STAT3	2	1	1	1	1	1.0
	LIG_SH2_STAT5	ELME000182	STAT5	3	3	2	3	3	1.0
	LIG_SH3_1	ELME000005		0	1	0	0	0	0.2
	LIG_SH3_3	ELME000155	SH3	1	2	1	1	1	1.0
	LIG_SH3_4	ELME000156		0	0	0	1	0	0.2
	LIG_SxIP_EBH_1	ELME000254	EBH	1	1	1	1	2	1.0
	LIG_TRAF2_1	ELME000117	TRAF2	1	1	0	1	1	0.8
	LIG_TYR_ITIM	ELME000020	ITIM	1	1	1	1	1	1.0
	LIG_Vh1_VBS_1	ELME000438		0	0	0	0	1	0.2
	LIG_WD40_WDR5_VDV_2	ELME000365	WDR5	2	10	3	8	9	1.0
	LIG_WW_3	ELME000135		0	1	0	0	0	0.2
Modification	MOD_CDK_SPK_2	ELME000429		0	0	1	0	0	0.2
	MOD_CDK_SPxK_1	ELME000153		0	1	0	0	0	0.2
	MOD_CK1_1	ELME000063	CK1-P	1	3	2	4	4	1.0
	MOD_CK2_1	ELME000064		3	3	1	0	0	0.6
	MOD_Cter_Amidation	ELME000093		1	0	0	0	0	0.2
	MOD_GlcNHglycan	ELME000085	GlyNH	1	2	3	1	2	1.0
	MOD_GSK3_1	ELME000053	GSK3-P	3	4	2	1	2	1.0
	MOD_NEK2_1	ELME000336	NEK2-P	2	3	3	2	4	1.0
	MOD_NEK2_2	ELME000337		0	0	1	0	0	0.2
	MOD_N-GLC_1	ELME000070		1	0	1	2	0	0.6
	MOD_PIKK_1	ELME000202		5	0	1	0	0	0.4
	MOD_PKA_1	ELME000008		0	1	0	0	0	0.2
	MOD_PKA_2	ELME000062	PKA-P	1	0	1	2	2	0.8
	MOD_Plk_1	ELME000442	PLK-P	4	1	2	1	1	1.0
	MOD_Plk_4	ELME000444		1	0	1	0	0	0.4
	MOD_ProDKin_1	ELME000159	p38-P	3	6	1	3	2	1.0
	MOD_SUMO_rev_2	ELME000393		1	1	0	0	0	0.4
Target	TRG_ENDOCYTIC_2	ELME000120		2	4	2	1	1	1.0
	TRG_Pf-PMV_PEXEL_1	ELME000462		1	2	1	1	1	1.0

GenBank accession numbers for betaretroviral integrase sequences are as follows: Endogenous retrovirus-K (ERVK; P10266.2), Exogenous mouse mammary tumor virus (MMTV; AAF31469.1), Mason–Pzifer monkey virus 5 (M-PMV; BBG56792.1), Enzootic nasal tumor virus (ENTV; ANG58699.1) and Jaagsiekte sheep retrovirus (JSRV; NP_041186.1). Asterisk indicates ERVK-specific ELM motif in Figure 1.

**Table 2 microorganisms-09-01509-t002:** ELM motifs in ERVK integrase and similar endogenous integrases in eukaryotes.

ELM Motif		ELM Accession	ERVK Integrase (*Homo sapiens*)	ERVK-8 Pol protein-Like (*Macaca fascicularis)*	Pol Protein (*Chlorocebus sabaeus*)	Pol Protein (*Fukomys darmarensis*)	Putative Protein (*Ochonta princeps*)	ERVK-8 pol Protein-Like (*Equus asinus*)	ERVK-18 pol Protein-Like (*Capra hircus*)	Pol Protein (*Ovis aries*)	Putative Protein (*Zonotrichia albicollis*)	Putative Protein (*Zosterops borbonicus*)	Integrase (*Onchocerca flexuosa*)	Motif conservation
Cleavage	CLV_C14_Caspase3-7	ELME000321	1	0	0	0	1	0	0	0	0	0	0	0.2
	CLV_NRD_NRD_1	ELME000102	0	0	0	0	0	0	0	1	0	0	1	0.2
	CLV_PCSK_KEX2_1	ELME000108	1	2	0	0	0	0	0	0	0	1	0	0.3
	CLV_PCSK_PC1ET2_1	ELME000100	1	2	0	0	0	0	0	0	0	1	0	0.3
	CLV_PCSK_SKI1_1	ELME000146	5	3	4	3	7	8	0	0	3	5	2	0.8
Degradation	DEG_APCC_DBOX_1	ELME000231	0	0	0	0	0	0	0	0	0	0	1	0.1
	DEG_MDM2_SWIB_1	ELME000184	0	0	0	0	0	0	0	0	1	0	0	0.1
Docking	DOC_CKS1_1	ELME000358	0	1	0	0	2	0	0	0	1	2	1	0.5
	DOC_CYCLIN_RxL_1	ELME000106	2	1	2	0	2	2	0	0	3	3	1	0.7
	DOC_MAPK_DCC_7	ELME000433	0	0	0	1	0	0	0	0	0	0	1	0.2
	DOC_MAPK_gen_1	ELME000233	0	1	0	3	5	2	1	1	2	4	1	0.8
	DOC_MAPK_HePTP_8	ELME000434	0	0	0	0	0	0	0	0	0	0	1	0.1
	DOC_MAPK_MEF2A_6	ELME000432	1	1	0	0	2	1	1	1	3	4	3	0.8
	DOC_PP1_RVXF_1	ELME000137	3	3	1	1	0	0	1	1	0	1	1	0.7
	DOC_PP2A_B56_1	ELME000425	0	0	0	1	0	0	0	0	0	0	0	0.1
	DOC_PP2B_LxvP_1	ELME000367	1	1	2	0	0	0	0	0	1	1	1	0.5
	DOC_PP2B_PxIxI_1	ELME000237	0	0	0	0	1	0	0	0	0	0	0	0.1
	DOC_PP4_FxxP_1	ELME000477	0	1	1	1	1	1	1	0	0	0	2	0.6
	DOC_USP7_MATH_1	ELME000239	1	3	0	1	1	1	2	2	3	1	3	0.9
	DOC_USP7_MATH_2	ELME000240	0	1	0	0	1	0	0	0	0	0	0	0.2
	DOC_USP7_UBL2_3	ELME000394	1	2	0	0	0	0	0	0	1	3	0	0.4
	DOC_WW_Pin1_4	ELME000136	3	0	1	0	6	2	2	2	3	3	1	0.8
Ligand	LIG_14-3-3_CanoR_1	ELME000417	2	4	1	1	2	2	1	2	3	1	1	1.0
	LIG_14-3-3_CterR_2	ELME000418	1	0	0	0	0	0	0	0	0	0	0	0.1
	LIG_Actin_WH2_2	ELME000313	0	0	1	1	0	0	1	1	0	1	1	0.5
	LIG_APCC_ABBA_1	ELME000435	0	0	0	1	0	1	0	0	0	0	0	0.2
	LIG_BIR_II_1	ELME000285	1	1	1	1	1	1	1	1	1	1	1	1.0
	LIG_BIR_III_4	ELME000293	0	0	0	0	0	0	1	1	0	0	0	0.2
	LIG_BRCT_BRCA1_1	ELME000197	1	2	0	1	0	0	1	1	0	0	2	0.5
	LIG_BRCT_BRCA1_2	ELME000198	1	1	0	0	0	0	0	0	0	0	0	0.2
	LIG_CSL_BTD_1	ELME000410	1	0	1	1	1	1	0	0	0	0	0	0.5
	LIG_deltaCOP1_diTrp_1	ELME000459	0	0	0	0	0	0	0	0	1	0	0	0.1
	LIG_EH1_1	ELME000148	0	0	1	0	0	0	1	1	0	0	0	0.3
	LIG_eIF4E_1	ELME000317	0	0	0	0	1	0	0	0	0	0	1	0.2
	LIG_FHA_1	ELME000052	5	5	5	3	2	0	0	1	1	1	4	0.8
	LIG_FHA_2	ELME000220	1	1	0	1	3	2	0	0	0	1	0	0.5
	LIG_LIR_Apic_2	ELME000369	0	3	3	2	1	2	1	1	1	0	1	0.8
	LIG_LIR_Gen_1	ELME000368	2	1	1	1	2	1	0	1	1	1	0	0.8
	LIG_LIR_Nem_3	ELME000370	6	5	7	6	7	3	4	4	1	3	6	1.0
	LIG_LYPXL_S_1	ELME000294	0	0	0	0	0	0	0	0	0	0	1	0.1
	LIG_PCNA_PIPBox_1	ELME000140	0	0	1	0	0	0	0	0	0	0	0	0.1
	LIG_PCNA_yPIPBox_3	ELME000482	0	0	0	2	2	2	0	0	0	0	0	0.3
	LIG_Pex14_1	ELME000080	1	0	0	0	1	0	0	0	2	2	0	0.4
	LIG_Pex14_2	ELME000328	2	3	1	2	0	2	1	1	1	2	1	0.9
	LIG_PTB_Apo_2	ELME000122	0	0	0	2	1	0	1	1	1	1	0	0.5
	LIG_PTB_Phospho_1	ELME000095	0	0	0	1	1	0	1	1	0	0	0	0.4
	LIG_REV1ctd_RIR_1	ELME000450	0	0	0	0	0	0	0	0	1	0	0	0.1
	LIG_RPA_C_Fungi	ELME000382	0	0	0	0	0	0	1	1	0	1	0	0.3
	LIG_SH2_CRK	ELME000458	0	1	0	1	2	2	0	0	1	1	0	0.5
	LIG_SH2_GRB2like	ELME000084	0	0	0	0	0	0	0	0	0	0	1	0.1
	LIG_SH2_NCK_1	ELME000474	0	1	0	1	0	1	0	0	1	1	0	0.5
	LIG_SH2_PTP2	ELME000083	1	0	1	1	1	0	1	1	1	0	1	0.7
	LIG_SH2_SRC	ELME000081	1	0	2	1	1	1	1	1	1	0	2	0.8
	LIG_SH2_STAP1	ELME000465	2	1	0	0	1	1	0	0	1	1	1	0.6
	LIG_SH2_STAT3	ELME000163	2	2	1	0	1	1	1	1	0	0	1	0.7
	LIG_SH2_STAT5	ELME000182	3	1	3	2	3	2	3	3	3	2	5	1.0
	LIG_SH3_3	ELME000155	1	2	1	1	3	2	1	1	5	4	2	1.0
	LIG_SUMO_SIM_par_1	ELME000333	0	1	0	1	0	0	0	0	1	2	0	0.4
	LIG_SxIP_EBH_1	ELME000254	1	0	1	2	0	1	0	2	0	1	1	0.6
	LIG_TRAF2_1	ELME000117	1	1	0	1	1	2	1	1	0	0	0	0.6
	LIG_TRAF6	ELME000133	0	0	0	0	0	1	0	0	0	0	0	0.1
	LIG_TRFH_1	ELME000249	0	0	0	0	0	0	0	0	0	0	1	0.1
	LIG_TYR_ITIM	ELME000020	1	0	2	1	1	0	1	1	1	0	1	0.7
	LIG_UBA3_1	ELME000395	0	0	1	0	0	0	0	0	0	0	1	0.2
	LIG_Vh1_VBS_1	ELME000438	0	0	0	1	0	0	1	1	0	1	0	0.4
	LIG_WD40_WDR5_VDV_1	ELME000364	0	0	0	0	0	0	0	0	0	1	0	0.1
	LIG_WD40_WDR5_VDV_2	ELME000365	2	10	6	5	6	8	8	8	10	10	12	1.0
	LIG_WW_3	ELME000135	0	0	0	1	0	0	0	0	0	0	0	0.1
Modification	MOD_CDK_SPK_2	ELME000429	0	0	1	0	1	0	0	0	0	0	1	0.3
	MOD_CDK_SPxxK_3	ELME000428	0	0	0	0	1	1	0	0	1	1	0	0.4
	MOD_CK1_1	ELME000063	1	0	1	2	3	1	3	3	4	3	3	0.9
	MOD_CK2_1	ELME000064	3	2	1	4	3	2	0	0	0	1	0	0.6
	MOD_CMANNOS	ELME000160	0	0	0	0	1	0	0	0	0	0	0	0.1
	MOD_Cter_Amidation	ELME000093	1	1	0	0	0	0	0	0	0	0	0	0.2
	MOD_GlcNHglycan	ELME000085	1	3	3	3	2	0	2	2	4	2	5	0.9
	MOD_GSK3_1	ELME000053	3	1	3	4	2	2	1	1	0	4	6	0.9
	MOD_NEK2_1	ELME000336	2	3	3	2	2	1	4	4	1	2	3	1.0
	MOD_NEK2_2	ELME000337	0	0	1	1	1	1	0	0	1	0	1	0.5
	MOD_N-GLC_1	ELME000070	3	0	0	2	1	1	0	0	2	1	1	0.6
	MOD_N-GLC_2	ELME000079	0	0	1	0	2	0	0	0	0	0	1	0.3
	MOD_PIKK_1	ELME000202	5	3	2	1	2	0	0	0	2	1	0	0.6
	MOD_PK_1	ELME000065	0	2	0	0	0	1	0	0	0	0	1	0.3
	MOD_PKA_1	ELME000008	0	1	0	1	0	1	0	0	0	1	0	0.4
	MOD_PKA_2	ELME000062	1	1	1	1	1	2	0	2	1	0	2	0.8
	MOD_Plk_1	ELME000442	4	2	1	2	2	4	1	1	2	3	2	1.0
	MOD_Plk_4	ELME000444	1	2	3	2	2	2	0	0	1	2	0	0.7
	MOD_ProDKin_1	ELME000159	3	3	1	0	6	2	2	2	3	3	1	0.9
	MOD_SUMO_for_1	ELME000002	0	0	0	0	0	0	0	0	2	0	0	0.1
	MOD_SUMO_rev_2	ELME000393	1	1	0	0	0	0	0	0	0	0	0	0.2
Targeting	TRG_ENDOCYTIC_2	ELME000120	2	1	3	2	2	2	1	1	2	1	2	1.0
	TRG_LysEnd_APsAcLL_1	ELME000149	0	0	0	0	0	0	0	0	1	0	1	0.2
	TRG_NLS_MonoExtC_3	ELME000278	0	0	0	0	0	1	0	0	0	0	0	0.1
	TRG_Pf-PMV_PEXEL_1	ELME000462	1	1	1	1	1	1	1	1	1	1	1	1.0

GenBank accession numbers for endogenous integrase sequences are as follows: Endogenous retrovirus-K (ERVK in Homo sapiens; P10266.2), ERVK-8 pol protein-like (Macaca fascicularis; XP_015309771.1), Pol protein (Chlorocebus sabaeus; KFO35018.1), Pol protein (Fukomys darmarensis; BBC20786.1), Putative protein (Ochonta princeps; XP_012786727.1), ERVK-8 pol protein-like (Equus asinus; XP_014715024.1), ERVK-18 pol protein-like (Capra hircus; XP_017905435.1), Pol protein (Ovis aries; ABV71120.1), Putative protein (Zonotrichia albicollis; TRZ15504.1), Putative protein (Zosterops borbonicus; XP_014125095.1), Integrase (Onchocerca flexuosa; OZC05619.1).

## Data Availability

National Center for Biotechnology Information (NCBI; https://www.ncbi.nlm.nih.gov/, (accessed on 12 July 2021)) can be used to access sequences listed in the paper.

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
