# Peer review of "Predicted Cellular Interactors of the Endogenous Retrovirus-K Integrase Enzyme"

_microorganisms, 2021, doi:10.3390/microorganisms9071509_

Round 1
Reviewer 1 Report
The manuscript by Benoit et al concerns an in silico analysis of putative binding partners of the ERV-K integrase protein, with an extensive discussion of the results. As retroviral integrase proteins perform their function in complexes, searching for protein interactions is an important research topic. And since such integrases can be expressed through the remains of endogenous retroviruses, with unknown, but possible deleterious effects on cellular processes, analyzing putative ERV-K IN interactions makes biological and medical sense. Of course one would love to see some lab experiments to follow up on the in silico analysis, but I realize that those will be published in the next paper due to expected information overload in the current manuscript.
The paper is well written and easy to follow, so my remarks are minor:
– the ERV-K IN protein was selected from a protein database. Of course, there are many more ERV-K integrations in the human genome. Is anything known about the conservation of the motifs found in other ERV-K IN coding sequences? Or: how representative is the chosen IN protein for ERV-K in general? And: are several ERV-K IN proteins expressed in human cells, or only one?
– How well do these interactome predictions align with actual experiments? I guess there must be data from HIV IN?
– Is anything known about the expression patterns of ERV-K IN? Is expression associated with certain cell types, and if yes, has that any influence on possible interacting partners? Meaning: how biologically relevant are the protein interactions retrieved from the database search? I would expect an endogenous protein to be likely expressed in any type of cell, while exogenous IN would only be expressed in cell types that can be infected by the particular virus so that endogenous IN can contribute broadly to disease?
– ERV-K IN interacting proteins differ from those known for HIV, such as LEDGF/p75 not being bound by the former. It could be that the composition of the PIC is important for integration site preference. Is anything known about such preferences in betaretroviruses compared with lentiviruses?
Author Response
We thank the reviewers and editors for their thoughtful comments for improvement of our manuscript. We have performed the suggested modifications, addressed reviewers’ comments, and corrected the text and figures accordingly.
Specific Response to Reviewer #1:
The manuscript by Benoit et al concerns an in silico analysis of putative binding partners of the ERV-K integrase protein, with an extensive discussion of the results. As retroviral integrase proteins perform their function in complexes, searching for protein interactions is an important research topic. And since such integrases can be expressed through the remains of endogenous retroviruses, with unknown, but possible deleterious effects on cellular processes, analyzing putative ERV-K IN interactions makes biological and medical sense. Of course one would love to see some lab experiments to follow up on the in silico analysis, but I realize that those will be published in the next paper due to expected information overload in the current manuscript.
Response: Thank you for your generous comments. We agree with the reviewer that experimental follow-up is essential – we are working on it and expect to have follow-up papers submitted shortly.
The paper is well written and easy to follow, so my remarks are minor:
1) the ERV-K IN protein was selected from a protein database. Of course, there are many more ERV-K integrations in the human genome. Is anything known about the conservation of the motifs found in other ERV-K IN coding sequences? Or: how representative is the chosen IN protein for ERV-K in general? And: are several ERV-K IN proteins expressed in human cells, or only one?
1) Thank you for this comment. The ERVK sequence used as an integrase reference is listed below Table 1 and 2 (Endogenous retrovirus-K (ERVK; P10266.2). However, we have added this information to the methods section proper on line 85. A full list of ERVK integrases in the human genome with enzymatic potential and conserved motifs is described in our previous publication (reference #22). The ERVK-10 sequence was selected for analysis in this paper as it is representative of a functional ERVK integrase (PMID: 8627815). In ALS patients, ERVK RNA expression originates from loci with putatively active ERVK integrases (PMID: 21280084), suggesting that several ERVK loci may drive integrase protein expression in cells. We have addressed these concerns on lines 51-52 and line 86 of the text.
2) How well do these interactome predictions align with actual experiments? I guess there must be data from HIV IN?
2) Although we have not disclosed the information in this manuscript, we have experimentally had success validating some of the ERVK integrase protein-protein interactions predicted herein. This will be the subject of upcoming manuscripts in the near future.
3) Is anything known about the expression patterns of ERV-K IN? Is expression associated with certain cell types, and if yes, has that any influence on possible interacting partners? Meaning: how biologically relevant are the protein interactions retrieved from the database search? I would expect an endogenous protein to be likely expressed in any type of cell, while exogenous IN would only be expressed in cell types that can be infected by the particular virus so that endogenous IN can contribute broadly to disease?
3) In our unpublished experiments, we can observe ERVK integrase expression at the protein level in human brain tissues (specifically neurons and astrocytes), human PBMC, and several human cell lines (both basal expression and stimulated through cytokines).
4) ERV-K IN interacting proteins differ from those known for HIV, such as LEDGF/p75 not being bound by the former. It could be that the composition of the PIC is important for integration site preference. Is anything known about such preferences in betaretroviruses compared with lentiviruses?
4) The ERVK-10 integrase exhibits strand-transfer activity for ERVK, HIV and RSV LTR substrates, suggesting a relaxed substrate specificity (PMID: 8627815). To the authors’ knowledge, it remains unclear how PIC composition differs between betaretroviruses and lentiviruses. However, it has been demonstrated that a reconstituted ERVK virus (HERV-KCon) exhibits integration site selectivity for transcription units and gene-rich regions, similar to HIV (PMID: 19270161).
Reviewer 2 Report
The manuscript entitled "Predicted Cellular Interactors of the Endogenous Retrovirus-K Integrase Enzyme” by Benoit and co-authors focuses on the in silico analysis of integrase (IN) enzyme of the members of ERVK family of endogenous retroviruses. The authors compared eukaryotic linear motifs (ELM) within ERVK integrase and integrases of other betaretroviruses, including exogenous retroviruses, such as ENTV and JSRV; characterized ELM and identified their potential interacting molecules. They identified a high affinity of ERVK IN for BRCA1, the 14-3-3 proteins and DNA damage response proteins. The authors analyzed potential involvement of ERVK in cellular signaling pathways, and based on that, assessed diseases and pathways implicated in the ERVK integrase interactome. Indeed, while the IN of clinically relevant exogenous retroviruses, especially of HIV-1, is well characterized, the knowledge about ERV integrases is fragmentary and requires further investigation. Whereas the replication of these symbiotic retroviruses is associated with critical processes of cell differentiation, placentation, neural transmission, as well as with autoimmune, neurodegenerative diseases and, likely, with various types of cancer. This makes the manuscript relevant and important to understand the biological properties of ERVK family members and to assess involvement of this particular enzyme in ERVK pathogenesis, particularly HERV-K HML-2 subgroup. The provided bioinformatics analysis will contribute to design further molecular studies to investigate the interacting partners of HERVK integrase in context of the diseases and pathways related to HERV-K pathogenesis.
Overall, the manuscript is of interest, well-written and logical, and is supposed to be important for an audience of virologists, immunologists and cell biologists. The manuscript structure is clear and solid. In general, the study is technically correct, methodically rigorous, providing clear and precise results. Without diminishing the undoubted scientific value of this study, below I provide a few suggestions that might improve some points of the manuscript.
- In section 3.4.4 the authors consider predicted interaction of ERVK IN with specific cellular transport systems including the factors involved in regulation of microtubule dynamics, cargo internalization via clathrin-mediated endocytosis, actin dynamics, etc. It should be noted here that IN cellular localization is strongly associated with the nucleus. At least for human ERVKs the ability to form viral particles and re-infect cells is questionable and presence of the processed IN in any significant amount in the cytoplasm is unlikely. The indicated interaction may be important for animal ERVKs like MMTV, but not for HERV-K. On the other hand, interaction of IN domain of Gag-Pro-Pol precursor with the elements of cytoplasmic transport machinery is theoretically possible in the case of the diseases related to HERV-K upregulation. This point could be considered in the manuscript.
- In section 3.5.2 the authors suggest involvement of HERV-K IN in neuropathogenic properties of HML-2 in the context of ALS, Alzheimer’s and prion diseases. Based on the diagram in Fig. 4, the predicted implication of IN enzyme in neuropathogenesis is moderate. Pathogenic role of HML-2 Env, Rec and Np9 is discussed, whereas involvement of Gag-Pro-Pol products is doubtful. I guess the authors should underline secondary potential role of IN in these pathogenic changes and mention the major impact of Env protein.
- Lines 421-424: “Collectively, our results point to ERVK IN driving a pattern of pathology that, depending on cellular context, may lead to carcinogenesis, neurodegeneration or contribute to diabetic complications”. Based on the fact that ERVKs are symbionts in mammalian cells for millions of years and in some of them gag-pro-pol genes are still transcriptionally functional, do the authors have any suggestions regarding involvement of the IN protein or the precursor protein in normal cellular processes, non-pathogenic pathways or co-option with any cellular functions? Considering of these potential options would be interesting.
Author Response
We thank the reviewers and editors for their thoughtful comments for improvement of our manuscript. We have performed the suggested modifications, addressed reviewers’ comments, and corrected the text and figures accordingly.
Specific Response to Reviewer #2:
The manuscript entitled "Predicted Cellular Interactors of the Endogenous Retrovirus-K Integrase Enzyme” by Benoit and co-authors focuses on the in silico analysis of integrase (IN) enzyme of the members of ERVK family of endogenous retroviruses. The authors compared eukaryotic linear motifs (ELM) within ERVK integrase and integrases of other betaretroviruses, including exogenous retroviruses, such as ENTV and JSRV; characterized ELM and identified their potential interacting molecules. They identified a high affinity of ERVK IN for BRCA1, the 14-3-3 proteins and DNA damage response proteins. The authors analyzed potential involvement of ERVK in cellular signaling pathways, and based on that, assessed diseases and pathways implicated in the ERVK integrase interactome. Indeed, while the IN of clinically relevant exogenous retroviruses, especially of HIV-1, is well characterized, the knowledge about ERV integrases is fragmentary and requires further investigation. Whereas the replication of these symbiotic retroviruses is associated with critical processes of cell differentiation, placentation, neural transmission, as well as with autoimmune, neurodegenerative diseases and, likely, with various types of cancer. This makes the manuscript relevant and important to understand the biological properties of ERVK family members and to assess involvement of this particular enzyme in ERVK pathogenesis, particularly HERV-K HML-2 subgroup. The provided bioinformatics analysis will contribute to design further molecular studies to investigate the interacting partners of HERVK integrase in context of the diseases and pathways related to HERV-K pathogenesis.
Overall, the manuscript is of interest, well-written and logical, and is supposed to be important for an audience of virologists, immunologists and cell biologists. The manuscript structure is clear and solid. In general, the study is technically correct, methodically rigorous, providing clear and precise results. Without diminishing the undoubted scientific value of this study, below I provide a few suggestions that might improve some points of the manuscript.
Response: Thank you for your generous comments. We agree with the reviewer that experimental follow-up is essential – we are working on it and expect to have follow-up papers submitted shortly.
1) In section 3.4.4 the authors consider predicted interaction of ERVK IN with specific cellular transport systems including the factors involved in regulation of microtubule dynamics, cargo internalization via clathrin-mediated endocytosis, actin dynamics, etc. It should be noted here that IN cellular localization is strongly associated with the nucleus. At least for human ERVKs the ability to form viral particles and re-infect cells is questionable and presence of the processed IN in any significant amount in the cytoplasm is unlikely. The indicated interaction may be important for animal ERVKs like MMTV, but not for HERV-K. On the other hand, interaction of IN domain of Gag-Pro-Pol precursor with the elements of cytoplasmic transport machinery is theoretically possible in the case of the diseases related to HERV-K upregulation. This point could be considered in the manuscript.
1) ERVK has exogenous retroviral origins. It is therefore possible that the ancestral exogenous ERVK precursor contained motifs permitting interaction with cellular transport systems in order to traverse the cell and mediate infection. It is as of yet unclear how an endogenous retrovirus could continue to interact with and impact these transport systems. We have now addressed this point on lines 336-338 of the text.
2) In section 3.5.2 the authors suggest involvement of HERV-K IN in neuropathogenic properties of HML-2 in the context of ALS, Alzheimer’s and prion diseases. Based on the diagram in Fig. 4, the predicted implication of IN enzyme in neuropathogenesis is moderate. Pathogenic role of HML-2 Env, Rec and Np9 is discussed, whereas involvement of Gag-Pro-Pol products is doubtful. I guess the authors should underline secondary potential role of IN in these pathogenic changes and mention the major impact of Env protein.
2) While we agree with the reviewer that to date the published literature points to a role for the ERVK env gene in neuropathogenesis, emerging data will show that ERVK IN is also a pathogenic driver of neurological disease. We are excited to share our findings on this topic in the near future.
3) Lines 421-424: “Collectively, our results point to ERVK IN driving a pattern of pathology that, depending on cellular context, may lead to carcinogenesis, neurodegeneration or contribute to diabetic complications”. Based on the fact that ERVKs are symbionts in mammalian cells for millions of years and in some of them gag-pro-pol genes are still transcriptionally functional, do the authors have any suggestions regarding involvement of the IN protein or the precursor protein in normal cellular processes, non-pathogenic pathways or co-option with any cellular functions? Considering of these potential options would be interesting.
3) Good point. We agree with the reviewer that beyond any potentially pathogenic impacts of ERVK IN, there could also be unexpected effects on normal cellular processes and physiology. One such possibility is DNA damage mediated lifespan extension (PMID: 26477511, 18978832). In the face of genomic insults, and when DNA damage pathways remain intact and select DDR proteins are overexpressed, it can have positive impacts on lifespan and stress resistance. These effects can also differ based on cell type specific DDR responses and sex of the host. Therefore the physiological impacts of ERVK IN (either pathological or beneficial) may be contextual on cell-type specific expression patterns and genetic background of the host. We have now addressed this point on lines 465-467 of the text.